# HYBRID DISCRIMINATIVE-GENERATIVE TRAINING VIA CONTRASTIVE LEARNING

## ABSTRACT

Contrastive learning and supervised learning have both seen significant progress and success. However, thus far they have largely been treated as two separate objectives, brought together only by having a shared neural network. In this paper we show that through the perspective of hybrid discriminative-generative training of energy-based models we can make a direct connection between contrastive learning and supervised learning. Beyond presenting this unified view, we show our specific choice of approximation of the energy-based loss significantly improves energy-based models and contrastive learning based methods in confidence-calibration, out-of-distribution detection, adversarial robustness, generative modeling, and image classification tasks. In addition to significantly improved performance, our method also gets rid of SGLD training and does not suffer from training instability. Our evaluations also demonstrate that our method performs better than or on par with state-of-the-art hand-tailored methods in each task.

## 1 INTRODUCTION

In the past few years, the field of deep learning has seen significant progress. Example successes include large-scale image classification (He et al., 2016; Simonyan & Zisserman, 2014; Srivastava et al., 2015; Szegedy et al., 2016) on the challenging ImageNet benchmark (Deng et al., 2009). The common objective for solving supervised machine learning problems is to minimize the cross-entropy loss, which is defined as the cross entropy between a target distribution and a categorical distribution called Softmax which is parameterized by the model's real-valued outputs known as logits. The target distribution usually consists of one-hot labels. There has been a continuing effort on improving upon the cross-entropy loss, various methods have been proposed, motivated by different considerations (Hinton et al., 2015; Müller et al., 2019; Szegedy et al., 2016).

Recently, contrastive learning has achieved remarkable success in representation learning. Contrastive learning allows learning good representations and enables efficient training on downstream tasks, an incomplete list includes image classification (Chen et al., 2020a;b; Grill et al., 2020; He et al., 2019; Tian et al., 2019; Oord et al., 2018), video understanding (Han et al., 2019), and knowledge distillation (Tian et al., 2019). Many different training approaches have been proposed to learn such representations, usually relying on visual pretext tasks. Among them, state-of-the-art contrastive methods (He et al., 2019; Chen et al., 2020a;c) are trained by reducing the distance between representations of different augmented views of the same image ('positive pairs'), and increasing the distance between representations of augment views from different images ('negative pairs').

Despite the success of the two objectives, they have been treated as two separate objectives, brought together only by having a shared neural network.

In this paper, to show a direct connection between contrastive learning and supervised learning, we consider the energy-based interpretation of models trained with cross-entropy loss, building on Grathwohl et al. (2019). We propose a novel objective that consists of a term for the conditional of the label given the input (the classifier) and a term for the conditional of the input given the label. We optimize the classifier term the normal way. Different from Grathwohl et al. (2019), we approximately optimize the second conditional over the input with a contrastive learning objective instead of a Monte-Carlo sampling-based approximation. In doing so, we provide a unified view on existing practice.

Our work takes inspiration from the work by Ng & Jordan (2002). In their 2002 paper, Ng & Jordan (2002) showed that classifiers trained with a generative loss (i.e., optimizing $p(x|y)$, with $x$ the input and $y$ the classification label) can outperform classifiers with the same expressiveness trained with a discriminative loss (i.e., optimizing $p(y|x)$). Later it was shown that hybrid discriminative generative model training can get the best of both worlds (Raina et al., 2004). The work by Ng & Jordan (2002) was done in the (simpler) context of Naive Bayes and Logistic Regression. Our work can be seen as lifting this work into today's context of training deep neural net classifiers.

Our empirical evaluation shows our method improves both the confidence-calibration and the classification accuracy of the learned classifiers, beating state-of-the-art methods. Despite its simplicity, our method outperforms competitive baselines in out-of-distribution (OOD) detection for all tested datasets. On hybrid generative-discriminative modeling tasks (Grathwohl et al., 2019), our method obtains superior performance without needing to run computational expensive SGLD steps. Our method learns significantly more robust classifiers than supervised training and achieves highly competitive results with hand-tailored adversarial robustness algorithms.

The contributions of this paper can be summarized as: (i) To the best of our knowledge, we are the first to reveal the connection between contrastive learning and supervised learning. We connect the two objectives through energy-based model. (ii) Built upon the insight, we present a novel framework for hybrid generative discriminative modeling via contrastive learning. (iii) Our method gets rid of SGLD therefore does not suffer from training instability of energy-based model. We empirically show that our method improves confidence-calibration, OOD detection, adversarial robustness, generative modeling, and classification accuracy, performing on par with or better than state-of-the-art energy-based models and contrastive learning algorithms for each task.

## 2 RELATED WORK

Our work falls into the category of hybrid generative discriminative models. Ng & Jordan (2002); Raina et al. (2004); Lasserre et al. (2006); Larochelle & Bengio (2008); Tu (2007); Lazarow et al. (2017) compare and study the connections and differences between discriminative model and generative model, and shows hybrid generative discriminative models can outperform purely discriminative models and purely generative models. Our work differs in that we propose an effective training approach in the context of deep neural network. By using contrastive learning to optimize the generative models, our method achieves state-of-the-art performance on a wide range of tasks.

Energy-based models (EBMs) have been shown can be derived from classifiers in supervised learning in the work of Xie et al. (2016); Du & Mordatch (2019), they reinterpret the logits to define a class-conditional EBM $p(x|y)$. Our work builds heavily on JEM (Grathwohl et al., 2019) which reveals that one can re-interpret the logits obtained from classifiers to define EBM $p(x)$ and $p(x, y)$, and shows this leads to significant improvement in OOD detection, calibration, and robustness while retain compelling classification accuracy. Our method differs in that we optimize our generative term via contrastive learning, buying the performance of state-of-the-art canonical EBMs algorithms (Grathwohl et al., 2019) without suffering from running computational expensive and slow SGLD (Welling & Teh, 2011) at every iteration.

Concurrent to our work, Winkens et al. (2020) proposes to pretrain using contrastive loss and then finetune with a joint supervised and contrastive loss, and shows the SimCLR loss improves likelihood-based OOD detection. Tack et al. (2020) also demonstrate contrastive learning improves OOD detection and calibration. Our work differs in that instead of a contrastive representation pre-train followed by supervised loss fine-tune, we use the contrastive loss to approximate a hybrid discriminative-generative model. We also empirically demonstrate our method enjoys broader usage by applying it to generative modeling, calibration, and adversarial robustness.

## 3 BACKGROUND

### 3.1 SUPERVISED LEARNING

In supervised learning, given a data distribution $p(x)$ and a label distribution $p(y|x)$ with $C$ categories, a classification problem is typically addressed using a parametric function, $f_\theta : \mathbb{R}^D \to \mathbb{R}^C$, which

maps each data point $x \in \mathbb{R}^D$ to $C$ real-valued numbers termed as logits. These logits are used to parameterize a categorical distribution using the Softmax function:

$$q_\theta(y|x) = \frac{\exp(f_\theta(x)[y])}{\sum_{y'} \exp(f_\theta(x)[y'])}, \tag{1}$$

where $f_\theta(x)[y]$ indicates the $y^{\text{th}}$ element of $f_\theta(x)$, $i.e.$, the logit corresponding to the $y^{\text{th}}$ class label. One of the most widely used loss functions for learning $f_\theta$ is minimizing the negative log likelihood:

$$\min_\theta -\mathbb{E}_{p_{\text{data}}(x,y)} \left[ \log q_\theta(y|x) \right]. \tag{2}$$

This loss function is often referred to as the cross-entropy loss function, because it corresponds to minimizing the KL-divergence with a target distribution $p(y|x)$, which consists of one-hot vectors with the non-zero element denoting the correct prediction.

## 3.2 ENERGY-BASED MODELS

**Energy-based models.**   Energy based models (EBMs) (LeCun et al., 2006) are based on the observation that probability densities $p(x)$ for $x \in \mathbb{R}^D$ can be expressed as

$$p_\theta(x) = \frac{\exp(-E_\theta(x))}{Z(\theta)}, \tag{3}$$

where $E_\theta(x) : \mathbb{R}^D \to \mathbb{R}$ maps each data point to a scalar; and $Z(\theta) = \sum_{x \in \mathcal{X}} \exp(-E_\theta(x))$ (or, for continuous $x$ we'd have $Z(\theta) = \int_{x \in \mathcal{X}} \exp(-E_\theta(x))$) is the normalizing constant, also known as the partition function. Here $\mathcal{X}$ is the full domain of $x$. For example, in the case of (let's say) 16x16 RGB images, computing $Z$ exactly would require a summation over $(256 \times 256 \times 256)^{(16 \times 16)} \approx 10^{2500}$ terms.

We can parameterize an EBM using any function that takes $x$ as the input and returns a scalar. For most choices of $E_\theta$, one cannot compute or even reliably estimate $Z(\theta)$, which means estimating the normalized densities is intractable and standard maximum likelihood estimation of the parameters, $\theta$, is not straightforward.

**Training EBMs.**   The log-likelihood objective for an EBM consists of a sum of $\log p_\theta(x)$ terms, one term for each data point $x$. The gradient of each term is given by:

$$\frac{\partial \log p_\theta(x)}{\partial \theta} = \mathbb{E}_{p_\theta(x')} \left[ \frac{\partial E_\theta(x')}{\partial \theta} \right] - \frac{\partial E_\theta(x)}{\partial \theta}, \tag{4}$$

where the expectation is over the model distribution $p_\theta(x')$. This expectation is typically intractable (for much the same reasons computing $Z(\theta)$ is typically intractable). However, it can be approximated through samples–assuming we can sample from $p_\theta$. Generating exact samples from $p_\theta$ is typically expensive, but there are some well established approximate (sometimes exact in the limit) methods based on MCMC (Grathwohl et al., 2019; Du & Mordatch, 2019; Hinton, 2002).

Among such sampling methods, recent success in training (and sampling from) energy-based models often relies on the Stochastic Gradient Langevin Dynamics (SGLD) approach (Welling & Teh, 2011), which generates samples by following this stochastic process:

$$x_0 \sim p_0(x), \qquad x_{i+1} = x_i - \frac{\alpha}{2} \frac{\partial E_\theta(x_i)}{\partial x_i} + \epsilon, \qquad \epsilon \sim \mathcal{N}(0, \alpha) \tag{5}$$

where $\mathcal{N}(0, \alpha)$ is the normal distribution with mean of 0 and standard deviation of $\alpha$, and $p_0(x)$ is typically a Uniform distribution over the input domain and the step-size $\alpha$ should be decayed following a polynomial schedule. The SGLD sampling steps are tractable, assuming the gradient of the energy function can be computed with respect to $x$, which is often the case. It is worth noting this process does *not* require evaluation the partition function $Z(\theta)$ (or any derivatives thereof).

**Joint Energy Models.**   The joint energy based model (JEM) (Grathwohl et al., 2019) shows that classifiers in supervised learning are secretly also energy-based based models on $p(x, y)$. The key

insight is that the logits $f_\theta(x)[y]$ in the supervised cross-entropy loss can be seen as defining an energy-based model over $(x, y)$, as follows:

$$p(x, y) = \frac{\exp(f_\theta(x)[y])}{Z(\theta)}, \tag{6}$$

where $Z(\theta)$ is the unknown normalization constant. I.e., matching this with the typical EBM notation, we have $f_\theta(x)[y] = -E_\theta(x, y)$. Subsequently, the density model of data points $p(x)$ can be obtained by marginalizing over $y$:

$$p(x) = \frac{\sum_y \exp(f_\theta(x)[y])}{Z(\theta)}, \tag{7}$$

with the energy $E_\theta(x) = -\log \sum_y \exp(f_\theta(x)[y])$. JEM (Grathwohl et al., 2019) adds the marginal log-likelihood $p(x)$ to the training objective, where $p(x)$ is expressed with the energy based model from Equation (7). JEM uses SGLD sampling for training.

### 3.3 CONTRASTIVE LEARNING

In contrastive learning (Hadsell et al., 2006; Gutmann & Hyvärinen, 2010; 2012; Mnih & Kavukcuoglu, 2013; Mikolov et al., 2013), it is common to optimize an objective of the following form:

$$\min_\theta -\mathbb{E}_{p_{\text{data}}(x)} \left[ \log \frac{\exp(h_\theta(x)^\top h_\theta(x'))}{\sum_{i=1}^K \exp(h_\theta(x)^\top h_\theta(x_i))} \right], \tag{8}$$

where $x$ and $x'$ are two different augmented views of the same data point, $K$ is the number of negative examples, $h_\theta : \mathbb{R}^D \to \mathbb{R}^H$ maps each data point to a normalized representation space with dimension $H$. This objective tries to maximally distinguish an input $x_i$ from alternative inputs $x'_i$. The intuition is that by doing so, the representation captures important information between similar data points, and therefore might improve performance on downstream tasks. This is usually called the contrastive learning loss or InfoNCE loss (Oord et al., 2018) and has been successful used for learning unsupervised representations (Sohn, 2016; Wu et al., 2018; He et al., 2019; Chen et al., 2020a). In the context of supervised learning, the Supervised Contrastive Loss (Khosla et al., 2020) shows that selecting $x_i$ from different categories as negative examples can improve the standard cross-entropy training. Their objective for learning the representation $h_\theta(x)$ is given by:

$$\min_\theta -\sum_{i=1}^{2N} \frac{1}{2N_{\tilde{y}_i} - 1} \sum_{j=1}^{2N} \mathbb{1}_{i \neq j} \mathbb{1}_{\tilde{y}_i = \tilde{y}_j} \log \frac{\exp(h_\theta(x_i)^\top h_\theta(x_j))}{\sum_{k=1}^{2N} \mathbb{1}_{i \neq k} \exp(h_\theta(x_i)^\top h_\theta(x_k))}, \tag{9}$$

where $N_{\tilde{y}_i}$ is the total number of images in the minibatch that have the same label $\tilde{y}_i$ as the anchor $i$. We'll see that our approach outperforms Supervised Contrastive Learning, while also simplifying by removing the need for selecting negative examples or pre-training a representation. Through the simplification we might get a closer hint at where the leverage is coming from.

## 4 HYBRID DISCRIMINATIVE GENERATIVE ENERGY-BASED MODEL (HDGE)

As in the typical classification setting, we assume we are given a dataset $(x, y) \sim p_{\text{data}}$. The primary goal is to train a model that can classify ($x$ to $y$). In addition, we would like the learned model to be capable of out-of-distribution detection, providing calibrated outputs, and serving as a generative model.

To achieve these goals, we propose to train a hybrid model, which consists of a discriminative conditional and a generative conditional by maximizing the sum of both conditional log-likelihoods:

$$\min_\theta -\mathbb{E}_{p_{\text{data}}(x,y)} \left[ \log q_\theta(y|x) + \log q_\theta(x|y) \right], \tag{10}$$

where $q_\theta(y|x)$ is a standard Softmax neural net classifier, and where $q_\theta(x|y) = \frac{\exp(f_\theta(x)[y])}{Z(\theta)}$, with $Z(\theta) = \sum_x \exp(f_\theta(x)[y])$.

The rationale for this objective originates from (Ng & Jordan, 2002; Raina et al., 2004), where they discuss the connections between logistic regression and naive Bayes, and show that hybrid discriminative and generative models can out-perform purely generative or purely discriminative counterparts. The main challenge with the objective from Equation (10) is the intractable partition function $Z(\theta)$.

Our main contribution is to propose a (crude, yet experimentally effective) approximation with a contrastive loss:

$$\mathbb{E}_{p_{\text{data}}(x,y)} \left[ \log q_\theta(x|y) \right] \tag{11}$$

$$= \mathbb{E}_{p_{\text{data}}(x,y)} \left[ \log \frac{\exp(f_\theta(x)[y])}{Z(\theta)} \right] \tag{12}$$

$$\approx \mathbb{E}_{p_{\text{data}}(x,y)} \left[ \log \frac{\exp(f_\theta(x)[y])}{\sum_{i=1}^{K} \exp(f_\theta(x_i)[y])} \right], \tag{13}$$

where $K$ denotes the number of normalization samples. This is similar to existing contrastive learning objectives, although in our formulation, we also use labels.

Intuitively, in order to have an accurate approximation in Equation (13), $K$ has to be sufficiently large—becoming exact in the limit of summing over all $x \in \mathcal{X}$. We don't know of any formal guarantees for our proposed approximation, and ultimately the justification has to come from our experiments. Nevertheless, there are two main intuitions we considered: (i) We try to make $K$ as large as is practical. Increasing $K$ is not trivial as it requires a larger memory. To still push the limits, following He et al. (2019) we use a memory bank to store negative examples. More specifically, we resort to using a queue to store past logits, and sample normalization examples from this queue during the training process. (ii) While in principle we would need to sum over all possible $x \in \mathcal{X}$, we could expect to achieve a good approximation by focusing on $(x, y)$ that have low energy. Since the training examples $x_i$ are encouraged to have low energy, we draw from those for our approximation. It is worth noting that the training examples $x_i, y_i$ are getting incorporated in the denominator using the same label $y$ as in the numerator. So effectively this objective is (largely) contrasting the logit value $f_\theta(x)[y]$ for $x$ with label $y$ from the logit values of other training examples $x_i$ that don't have the same label $y$.

To bring it all together, our objective can be seen as a hybrid combination of supervised learning and contrastive learning given by:

$$\min_\theta -\mathbb{E}_{p_{\text{data}}(x,y)} \left[ \alpha \log q_\theta(y|x) + (1-\alpha) \log q_\theta(x|y) \right] \tag{14}$$

$$\approx \min_\theta -\mathbb{E}_{p_{\text{data}}(x,y)} \left[ \alpha \log \frac{\exp(f_\theta(x)[y])}{\sum_{y'} \exp(f_\theta(x)[y'])} + (1-\alpha) \log \frac{\exp(f_\theta(x)[y])}{\sum_{i=1}^{K} \exp(f_\theta(x_i)[y])} \right], \tag{15}$$

where $\alpha$ is weight between $[0, 1]$. When $\alpha = 1$, the objective reduces to the standard cross-entropy loss, while $\alpha = 0$, it reduces to an end-to-end supervised version of contrastive learning. We evaluated these variants in experiments, and we found that $\alpha = 0.5$ delivers the highest performance on classification accuracy as well as robustness, calibration, and out-of-distribution detection.

The resulting model, dubbed Hybrid Discriminative Generative Energy-based Model (HDGE), learns to jointly optimize supervised learning and contrastive learning. A PyTorch (Paszke et al., 2019)-like pseudo code corresponding to this algorithm is included in Appendix Algorithm 1.

## 5 EXPERIMENT

### 5.1 OUT-OF-DISTRIBUTION DETECTION

We conduct experiments to evaluate HDGE on out-of-distribution (OOD) detection tasks. In general, OOD detection is a binary classification problem, where the model is required to produce a score $s_\theta(x) \in \mathbb{R}$, where $x$ is the query, and $\theta$ is the model parameters. We desire that the scores for in-distribution examples are higher than that out-of-distribution examples. Following the setting of Grathwohl et al. (2019), we use the area under the receiver-operating curve (AUROC) (Hendrycks & Gimpel, 2016) as the evaluation metric. In our evaluation, we will consider two different score functions, the input density $q(x)$ (Section 5.1.1) and the predictive distribution $q(y|x)$ (Section 5.1.2).

### 5.1.1 INPUT DENSITY $q(x)$

| $s_\theta(x)$ | Model | Out-of-distribution | | | |
|---|---|---|---|---|---|
| | | SVHN | Interp | CIFAR100 | CelebA |
| $\log q(x)$ | WideResNet-28-10 | .46 | .41 | .47 | .49 |
| | Unconditional Glow | .05 | .51 | .55 | .57 |
| | Class-Conditional Glow | .07 | .45 | .51 | .53 |
| | IGEBM | .63 | .70 | .50 | .70 |
| | JEM | .67 | .65 | .67 | .75 |
| | HDGE (ours) | **.96** | **.82** | **.91** | **.80** |
| | JointLoss(ResNet-50) | .995 | - | .929 | - |
| | HDGE (ResNet-50) | .997 | - | .938 | - |
| $\max_y p(y|x)$ | WideResNet-28-10 | .93 | **.77** | .85 | .62 |
| | Contrastive pretraining | .87 | .65 | .80 | .58 |
| | Class-Conditional Glow | .64 | .61 | .65 | .54 |
| | IGEBM | .43 | .69 | .54 | .69 |
| | JEM | .89 | .75 | **.87** | .79 |
| | HDGE (ours) | **.95** | .76 | .84 | **.81** |

Table 1: OOD Detection Results. The model is WideResNet-28-10 (without BN) following the settings of JEM (Grathwohl et al., 2019). The comparison with JointLoss (Winkens et al., 2020) follows their setting to use ResNet-50. The results of JointLoss are obtained from its paper. The training dataset is CIFAR-10. Values are AUROC. Standard deviations given in Table 4 (Appendix).

Prior work show that fitting a density model on the data and consider examples with low likelihood to be OOD is effective, and the likelihoods from EBMs can be reliably used as a predictor for OOD inputs (Du & Mordatch, 2019; Grathwohl et al., 2019). We are interested in whether HDGE results in better likelihood function for OOD detection. All the methods are based on the WideResNet-28-10 (Zagoruyko & Komodakis, 2016). We follow the same experiment settings of Grathwohl et al. (2019) to remove the batch normalization (BN) (Ioffe & Szegedy, 2015) in WideResNet-28-10. In addition to standard discriminative models and hybrid model JEM, we also compare HDGE with other canonical algorithms: 1) Glow (Kingma & Dhariwal, 2018) which is a compelling flow-based generative model. 2) JointLoss (Winkens et al., 2020), a recent state-of-the-art which proposes to pretrain using contrastive loss and then finetune with a joint supervised and contrastive loss, and shows the SimCLR loss improves likelihood-based OOD detection. The results are shown in Table 1 (top), HDGE consistently outperforms all of the baselines. The corresponding distribution of score are visualized in Figure 1, it shows that HDGE correctly assign lower scores to out-of-distribution samples and performs extremely well on detecting samples from SVHN, CIFAR-100, and CelebA.

Interestingly, while Nalisnick et al. (2019) demonstrates powerful neural generative models trained to estimate density $p(x)$ can perform poorly on OOD detection, often assigning higher scores to OOD data points (e.g. SVHN) than in-distribution data points (e.g. CIFAR10), HDGE successfully assign higher scores only to in-distribution data points as shown in the histograms in Figure 1.

We believe that the improvement of HDGE over JEM is due to compared with SGLD sampling based methods, HDGE holds the ability to incorporate a large number and diverse samples and their corresponding labels information to train the generative conditional $\log q(x|y)$. Comparing with contrastive learning approach (Winkens et al., 2020), HDGE differs in that the contrastive loss inside the $\log q(x|y)$ utilizes label information to help contrast similar data points. The empirical advantage of HDGE over JointLoss shows the benefit of incorporating label information.

### 5.1.2 PREDICTIVE DISTRIBUTION $p(y|x)$

A widely used OOD score function is the maximum prediction probability (Hendrycks & Gimpel, 2016) which is given by $s_\theta(x) = \max_y p_\theta(y|x)$. Intuitively, a model with high classification accuracy tends to has a better OOD performance using this score function.

We compare with HDGE with standard discriminative models, generative models, and hybrid models. We also evaluate a contrastive pre-training baseline which consists of learning a representation via contrastive learning and training a linear classifier on top of the representation.

The results of OOD detection are show in Table 5.1 (bottom). We find HDGE performs beyond the performance of a strong baseline classifier and considerably outperforms all other generative modeling and hybrid modeling methods. The OOD detection evaluation shows that it is helpful to jointly train the generative model $q(x|y)$ together with the classifier $p(y|x)$ to have a better classifier model. HDGE provides an effective and simple approach to improve out-of-distribution detection.

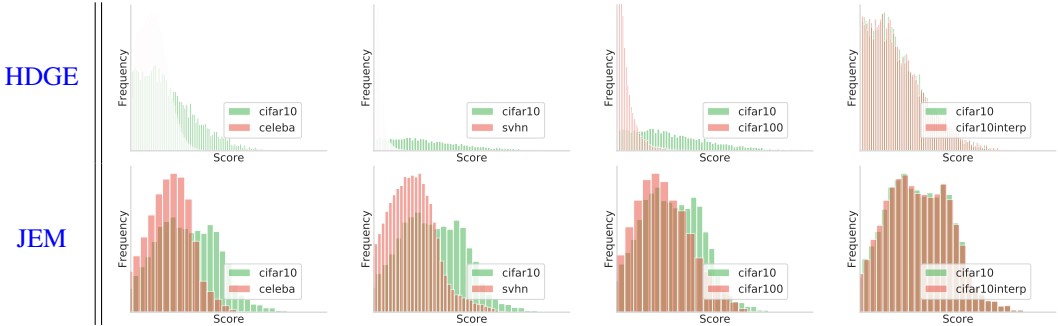

Figure 1: Histograms for OOD detection using density $q(x)$ as score function. The model is WideResNet-28-10 (without BN). Green corresponds to the score on (in-distribution) training dataset CIFAR-10, and red corresponds to the score on the testing dataset. The cifar10interp denotes a dataset that consists of a linear interpolation of the CIFAR-10 dataset.

## 5.2 CONFIDENCE-CALIBRATION

Calibration plays an important role when deploy the model in real-world scenarios where outputting an incorrect decision can have catastrophic consequences (Guo et al., 2017). The goodness of calibration is usually evaluated in terms of the Expected Calibration Error (ECE), which is a metric to measure the calibration of a classifier.

It works by first computing the confidence, $\max_y p(y|x_i)$, for each $x_i$ in some dataset and then grouping the items into equally spaced buckets $\{B_m\}_{m=1}^M$ based on the classifier's output confidence. For example, if $M = 20$, then $B_0$ would represent all examples for which the classifier's confidence was between 0.0 and 0.05. The ECE is defined as following:

$$\text{ECE} = \sum_{m=1}^M \frac{|B_m|}{n} |\text{acc}(B_m) - \text{conf}(B_m)|, \tag{16}$$

where $n$ is the number of examples in the dataset, $\text{acc}(B_m)$ is the averaged accuracy of the classifier of all examples in $B_m$ and $\text{conf}(B_m)$ is the averaged confidence over all examples in $B_m$. For a perfectly calibrated classifier, this value will be 0 for any choice of $M$. Following Grathwohl et al. (2019), we choose $M = 20$ throughout the experiments. A classifier is considered calibrated if its predictive confidence, $\max_y p(y|x)$, aligns with its misclassification rate. Thus, when a calibrated classifier predicts label $y$ with confidence score that is the same at the accuracy.

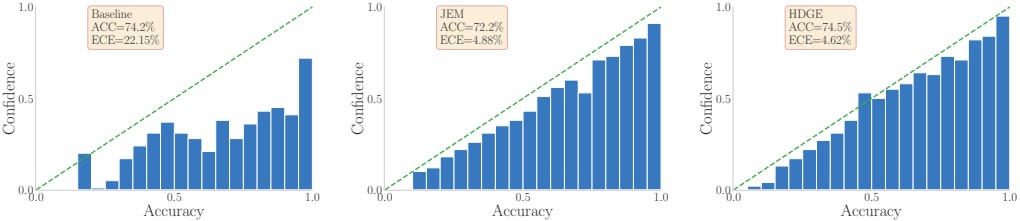

Figure 2: CIFAR-100 calibration results. The model is WideResNet-28-10 (without BN). Expected calibration error (ECE) (Guo et al., 2017) on CIFAR-100 dataset under various training losses.

We evaluate the methods on CIFAR-100 where we train HDGE and baselines of the same architecture, and compute the ECE on hold-out datasets. The histograms of confidence and accuracy of each method are shown in Figure 2.

While classifiers have grown more accurate in recent years, they have also grown considerably less calibrated (Guo et al., 2017), as shown in the left of Figure 2. Grathwohl et al. (2019) significantly

improves the calibration of classifiers by optimizing $q(x)$ as EBMs training (Figure 2 middle), however, their method is computational expensive due to the contrastive divergence and SGLD sampling process and their training also sacrifices the accuracy of the classifiers. In contrast, HDGE provides a computational feasible method to significantly improve both the accuracy and the calibration at the same time (Figure 2 right).

## 5.3 IMAGE CLASSIFICATION

We compare HDGE with (i) the supervised learning baseline uses the standard cross-entropy loss. We follow the settings of Zagoruyko & Komodakis (2016) for evaluation on CIFAR-10 and CIFAR-100, and we decay the learning rate by 0.2 at epoch 60, 120, 160. (ii) Supervised Contrastive Learning from (Khosla et al., 2020), which proposes to use label information to select negative examples at the contrastive pre-training stage, and shows incorporating the label information helps the downstream supervised training of classifiers. We adapt the official implementation of the Supervised Contrastive Loss to use WideResNet. (iii) JEM from (Grathwohl et al., 2019), which proposes to incorporate energy-based modeling training with the standard cross-entropy loss.

As reported in Table 2, HDGE outperforms standard Supervised Learning (which uses only the $q_\theta(y|x)$ loss term), outperforms Supervised Contrastive Learning from Khosla et al. (2020) (which uses a different approximation to the $q_\theta(y|x)$), outperforms JEM (which uses the classification loss on $q_\theta(y|x)$ supplemented with a loss on the marginal $q_\theta(x)$), and outperforms HDGE with $\log q_\theta(x|y)$ (which only trains the generative loss term). This shows the benefit of hybrid discriminative and generative model via jointly optimizing the discriminative (classifier) loss and the generative (contrastive) loss. In addition, when studying methods that only have the generative term $q_\theta(x|y)$, we see that HDGE ($\log q_\theta(x|y)$ only) achieves higher accuracy than Khosla et al. (2020), verifying our method provides an improved generative loss term.

| Dataset | Method | | | | |
|---------|----------------------|---------------------------|----------|-------------|----------------------------|
|  | Supervised Learning | Supervised Contrastive | JEM | HDGE (ours) | HDGE ($\log q_\theta(x|y)$ only) |
| CIFAR10 | $95.8 \pm .15$ | $96.3 \pm .24$ | $94.4 \pm .17$ | $\mathbf{96.7 \pm .10}$ | $96.4 \pm .12$ |
| CIFAR100 | $79.9 \pm .21$ | $80.5 \pm .21$ | $78.1 \pm .10$ | $\mathbf{80.9 \pm .09}$ | $80.6 \pm .10$ |

Table 2: Comparison on three standard image classification datasets: All models use the same batch size of 256 and step-wise learning rate decay, the number of training epochs is 200. The baselines Supervised Contrastive (Khosla et al., 2020), JEM (Grathwohl et al., 2019), and our method HDGE are based on WideResNet-28-10 (Zagoruyko & Komodakis, 2016).

## 5.4 HYBRID DISCRIMINATIVE-GENERATIVE MODELING TASKS

HDGE models can be sampled from with SGLD. However, during experiments we found that adding the marginal log-likelihood over $x$ (as done in JEM) improved the generation. we hypothesis that this is due the approximation via contrastive learning focuses on discriminating between images of different categories rather than estimating density.

So we evaluated generative modeling through SGLD sampling from a model trained with the following objective:

$$\min_\theta \mathbb{E}_{p_{\mathrm{data}}(x,y)} \left[ \log q_\theta(y|x) + \log q_\theta(x|y) + \log q_\theta(x) \right], \qquad (17)$$

where $\log q_\theta(x)$ is optimized by running SGLD sampling and contrastive divergence as in JEM and $\log q_\theta(y|x) + \log q_\theta(x|y)$ is optimized through HDGE.

We train this approach on CIFAR-10 and compare against other hybrid models as well as standalone generative and discriminative models. We present inception scores (IS) (Salimans et al., 2016) and Frechet Inception Distance (FID) (Heusel et al., 2017) given that we cannot compute normalized likelihoods. The results are shown in Table 3 and Figure 3.

The results show that jointly optimizing $\log q_\theta(y|x) + \log q_\theta(x|y) + \log q_\theta(x)$ by HDGE (first two terms) and JEM (third term) together can outperform optimizing $\log q_\theta(y|x) + \log q_\theta(x)$ by JEM, and

it significantly improves the generative performance over the state of the art in generative modeling methods and retains high classification accuracy simultaneously. We believe the superior performance of HDGE + JEM is due to the fact that HDGE learns a better classifier and JEM can exploit it and maybe optimizing $\log q(x|y)$ via HDGE is a good auxiliary objective.

| Class | Model | Accuracy% ↑ | IS↑ | FID↓ |
|-------|-------|-------------|-----|------|
| **Hybrid** | Residual Flow | 70.3 | 3.6 | 46.4 |
| | Glow | 67.6 | 3.92 | 48.9 |
| | IGEBM | 49.1 | 8.3 | 37.9 |
| | JEM | 92.9 | 8.76 | 38.4 |
| | HDGE | **94.6** | N/A | N/A |
| | HDGE+JEM | 94.4 | **9.19** | **37.6** |
| **Disc.** | WideResNet | 95.8 | N/A | N/A |
| | WideResNet(w/o BN) | 93.6 | N/A | N/A |
| **Gen.** | SNGAN | N/A | 8.59 | 25.5 |
| | NCSN | N/A | 8.91 | 25.32 |

Table 3: Hybrid modeling results on CIFAR-10. All models are based on WideResNet-28-10 (Zagoruyko & Komodakis, 2016)(without BN). Residual Flow (Chen et al., 2019), Glow (Kingma & Dhariwal, 2018), IGEBM (Du & Mordatch, 2019), SNGAN (Miyato et al., 2018), NCSN (Song & Ermon, 2019), JEM (Grathwohl et al., 2019)

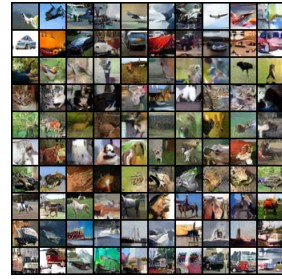

Figure 3: Class-conditional samples generated by running HDGE+JEM on CIFAR-10.

## 5.5 ADVERSARIAL ROBUSTNESS

The commonly considered adversarial attack is the $L_p$-norm constrained adversarial examples, which are defined as $\hat{x} \in B(x, \epsilon)$ that changes the model's prediction, where $B(x, r)$ denotes a ball centered at $x$ with radius $r$ under the $L_p$-norm metric. In this work, we run white-box PGD (projected gradient descent) attack with respect to the $L_2$ and $L_\infty$ norms, giving the attacker access to gradients, in which PGD is used to find a local maximal within a given perturbation ball (Madry et al., 2017). We train HDGE and compare with the state-of-the-art adversarial training methods. Adv Training (Madry et al., 2017; Santurkar et al., 2019) which proposes to use robust optimization to train classifier to be robust to the norm through which it is being attacked. Results from the PGD experiments can be seen in Figure 4. We can see that HDGE can achieve compelling robustness to the state-of-the-art adversarial training methods.

We note that while JEM improves the robustness too by optimizing the likelihood of EBMs, it requires computationally expensive SGLD sampling procedure. In contrast, HDGE significantly improves the robustness of standard classifiers by computationally scalable contrastive learning.

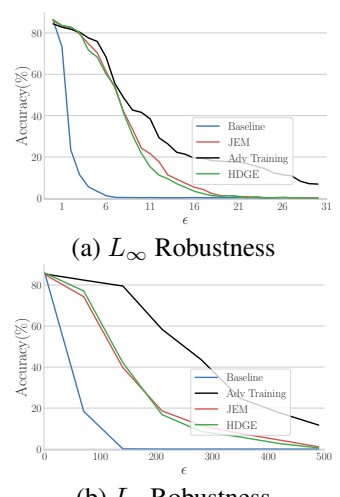

(a) $L_\infty$ Robustness

(b) $L_2$ Robustness

Figure 4: **Adversarial robustness** results with PGD attacks. HDGE adds considerable robustness to standard supervised training and achieves comparable robustness with JEM.

## 6 CONCLUSION

In this work, we develop HDGE, a new framework for supervised learning and contrastive learning through the perspective of hybrid discriminative and generative model. We propose to leverage contrastive learning to approximately optimize the model for discriminative and generative tasks. JEM (Grathwohl et al., 2019) shows energy-based models have improved confidence-calibration, out-of-distribution detection, and adversarial robustness. HDGE builds on top of JEM and contrastive learning beats JEM and contrastive loss in all of the tasks and performs significantly better or on par with state-of-the-art hand-tailored methods in each task. HDGE gets rid of SGLD therefore does not suffer from training instability and is also conceptual simple to implement. We hope HDGE will be useful for future research of hybrid discriminative-generative training.

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

## A  PYTORCH LIKE IMPLEMENTATION

---

**Algorithm 1** Hybrid Discriminative-Generative Training via Contrastive Learning

---

```
# f: encoder networks for images
# queue: queue of probabilities (KxC)
# B: batch size; K: queue size; C: number of
classes; m: momentum; t: temperature
for x, y in loader:  # load a minibatch with B
images x and labels y
    x = aug(x)   # optional random data
    augmentation
    y = t(y) # transform to one-hot vector with
    dimension C
    # logits: KxC
    ce_logits = f.forward(x)

    # standard cross-entropy loss
    # optimize log q(y|x)
    ce_loss = CrossEntropyLoss(ce_logits, y)

    # normalize logits by L2: BxC
    probs = normalize(ce_logits)

    # positive logits: Bx1
    l_pos = logsumexp(probs*y, dim=1,
    keepdim=True)
```

```
# negative logits: BxK
l_neg = einsum("bc,ck->bck", [y,
queue.clone().detach()])   # BxCxK
l_neg = logsumexp(l_neg, dim=1)   # BxK

# logits: Bx(1+K)
logits = cat([l_pos, l_neg], dim=1)

# positives are the 0-th
labels = zeros(K)

# contrastive loss
# optimize log q(x|y)
cl_loss = CrossEntropyLoss(logits/t, labels)

# hybrid training
loss = ce_loss + cl_loss

# SGD update: encoder network
loss.backward()
update(f.params)

# update buffer
enqueue(queue, probs)   # enqueue the current
minibatch of probs
dequeue(queue)   # dequeue the earliest
minibatch
```

---

einsum: Einstein sum; cat: concatenation; logsumexp: LogSumExp operation.

| $s_\theta(x)$ | Model | SVHN | Out-of-distribution | | |
| | | | Interp | CIFAR100 | CelebA |
|---|---|---|---|---|---|
| $\log p(x)$ | WideResNet-28-10 | $.46 \pm .21$ | $.41 \pm .19$ | $.47 \pm .23$ | $.49 \pm .21$ |
| | Unconditional Glow | $.05 \pm .01$ | $.51 \pm .23$ | $.55 \pm .22$ | $.57 \pm .19$ |
| | Class-Conditional Glow | $.07 \pm .02$ | $.45 \pm .21$ | $.51 \pm .19$ | $.53 \pm .17$ |
| | IGEBM | $.63 \pm .20$ | $.70 \pm .19$ | $.50 \pm .14$ | $.70 \pm .14$ |
| | JEM | $.67 \pm .11$ | $.65 \pm .14$ | $.67 \pm .15$ | $.75 \pm .12$ |
| | HDGE (ours) | $\mathbf{.96} \pm .08$ | $\mathbf{.82} \pm .11$ | $\mathbf{.91} \pm .09$ | $\mathbf{.80} \pm .12$ |
| | JEM + HDGE (ours) | $.95 \pm .12$ | $.82 \pm .13$ | $.90 \pm .12$ | $.80 \pm .15$ |
| | JointLoss(ResNet-50) | $.995 \pm .10$ | - | $.929 \pm .09$ | - |
| | HDGE (ResNet-50) | $.995 \pm .08$ | - | $.938 \pm .06$ | - |
| $\max_y p(y\vert x)$ | WideResNet-28-10 | $.93 \pm .13$ | $\mathbf{.77} \pm .11$ | $.85 \pm .21$ | $.62 \pm .23$ |
| | Contrastive pretraining | $.87 \pm .11$ | $.65 \pm .15$ | $.80 \pm .16$ | $.58 \pm .17$ |
| | Class-Conditional Glow | $.64 \pm .21$ | $.61 \pm .26$ | $.65 \pm .17$ | $.54 \pm .22$ |
| | IGEBM | $.43 \pm .13$ | $.69 \pm .21$ | $.54 \pm .16$ | $.69 \pm .19$ |
| | JEM | $.89 \pm .13$ | $.75 \pm .18$ | $\mathbf{.87} \pm .21$ | $.79 \pm .22$ |
| | HDGE (ours) | $\mathbf{.95} \pm .11$ | $.76 \pm .12$ | $.84 \pm .09$ | $\mathbf{.81} \pm .07$ |
| | JEM + HDGE (ours) | $.94 \pm .18$ | $.77 \pm .13$ | $.88 \pm .13$ | $.80 \pm .21$ |

Table 4: OOD Detection Results. The model is WideResNet-28-10 (without BN) following the settings of JEM (Grathwohl et al., 2019), except ResNet-50 when comparing with JointLoss (Winkens et al., 2020). The training dataset is CIFAR-10. Values are AUROC. Results of the baselines are from Grathwohl et al. (2019) and Winkens et al. (2020).

## B  EXPERIMENT DETAILS

### B.1  EVALUATION DATA

To evaluate HDGE, we completed a thorough empirical investigation on several standard datasets: CIFAR-10 and CIFAR-100 (Krizhevsky et al., 2009), two labeled datasets composed of $32 \times 32$ images with 10 and 100 classes respectively (Sections 5.1, 5.2, 5.3 and 5.4); SVHN (Netzer et al., 2011), a labeled dataset composed of over $600,000$ digit images (Section 5.1); CelebA (Liu et al., 2015), a labeled dataset consisting of over $200,000$ face images and each with $40$ attribute annotation (Section 5.1).

## B.2 Training details

Our training settings follow exactly that of JEM, except stated otherwise in some ablation study. Pseudo-code for our training procedure is in Algorithm 1.

The cross-entropy baseline is based on the code from the official PyTorch training code [1]. HDGE's implementation is based on the official codes of MoCo [2] and JEM [3]. Our source code in PyTorch (Paszke et al., 2019) is available online [4].

In the OOD evaluation, the results of JointLoss are obtained from Winkens et al. (2020), the results of JEM and other baselines are obtained from Grathwohl et al. (2019).

Our method HDGE follows the experimental settings of JEM and have exactly the same hyperparameters in optimization and model choices as JEM. the temperature $\tau = 0.1$ as in other experiments conducted in this work.

One baseline JointLoss (Winkens et al., 2020) uses a different model ResNet-50, to have a fair comparison, our HDGE (ResNet-50) also uses ResNet-50. JointLoss also incorporates multiple training techniques such as LARS optimizer and label smoothing to help training which we do not use in HDGE.

The likelihood score $\log q(x)$ is calculated by applying `LogSumExp` operation on the $\log q(y|x)$ within HDGE. Specifically,

$$\log q(x) = \log \sum_y q(x, y) = \log \sum_y \frac{\exp(f_{(x)}[y])}{Z}, \qquad (18)$$

where $Z$ is the normalization constant. The score $\log q(x)$ we care about is then $\sum_y \exp(f_{(x)}[y]) = -\texttt{LogSumExp}_y(f(x)[y])$. A similar scheme also proposed in recent OOD detection work (Liu et al., 2020).

## C   SimCLR style implementation of $\log p(x|y)$

We conducted a comparison between HDGE with MoCo and SimCLR style approximations of the contrastive loss in $\log p(x|y)$. One of the key differences between MoCo and SimCLR is that MoCo uses a gradient disablaed memory to save logits while SimCLR simply increase batch size. Chen et al. (2020a;b) demonstrate that SimCLR can outperform MoCo significantly.

We use batch size $2048$ in our SimCLR style HDGE and its pseudo code similar to Algorithm 1 is shown in Algorithm 2.

The results are shown in Table 5, we can see that SimCLR style of HDGE performs comparably with the default MoCo style, indicating Hybrid Discriminative-Generative Training is insensible to detail implementation choices. However, our default implementation Algorithm 1 has less requirements on computation memory size, which makes it widely applicable.

## D   Goodness of approximation

Since we made the approximation to energy-based model by contrastive learning in Equation (13), we are interested in evaluating the impact of the number of negative examples $K$ on the goodness of this approximation. We consider a classification task and a density based OOD detection task as proxies of evaluating the approximation.

**Classification.**   We compare the image classification of HDGE on CIFAR-100. The results are shown in Figure 5. We found that increasing number of negative samples $K$ improves the performance of HDGE, and with sufficient number of negative examples HDGE significantly outperform the

---

[1]https://github.com/szagoruyko/wide-residual-networks/tree/master/pytorch

[2]https://github.com/facebookresearch/moco

[3]https://github.com/wgrathwohl/JEM

[4]anonymous during double-blind review

---

**Algorithm 2** HDGE with SimCLR style approximation

---

```
# f: encoder networks for images
# B: batch size; C: number of classes; t:
temperature
for x, y in loader:  # load a minibatch with B
images x and labels y
    x = aug(x)   # random data augmentation
    y = t(y) # transform to one-hot vector with
dimension C
    # logits: BxC
    ce_logits = f.forward(x)
    # standard cross-entropy loss
    # optimize log q(y|x)
    ce_loss = CrossEntropyLoss(ce_logits, y)
    # normalize logits by L2: BxC
    probs = normalize(ce_logits)
    # positive logits: Bx1
```

```
    l_pos = logsumexp(probs*y, dim=1,
keepdim=True)
    # negative logits: BxB
    l_neg = einsum("bc,bc->bcb", [y, x])
    l_neg = logsumexp(l_neg, dim=1)
    # logits: Bx(1+B)
    logits = cat([l_pos, l_neg], dim=1)
    # positives are the 0-th
    labels = zeros(K)
    # contrastive loss
    # optimize log q(x|y)
    cl_loss = CrossEntropyLoss(logits/t, labels)
    # hybrid training
    loss = ce_loss + cl_loss
    # SGD update: encoder network
    loss.backward()
    update(f.params)
```

---

einsum: Einstein sum; cat: concatenation; logsumexp: LogSumExp operation.

| $s_\theta(x)$ | HDGE | | Out-of-distribution | | |
| | | SVHN | Interp | CIFAR100 | CelebA |
|---|---|---|---|---|---|
| $\log p(x)$ | default | $.96 \pm .08$ | $\mathbf{.82} \pm .11$ | $\mathbf{.91} \pm .09$ | $\mathbf{.80} \pm .12$ |
| | SimCLR style | $\mathbf{.97} \pm .06$ | $.82 \pm .13$ | $.91 \pm .11$ | $.80 \pm .15$ |
| | no logits normalization | $.96 \pm .12$ | $.82 \pm .11$ | $.91 \pm .09$ | $.80 \pm .12$ |
| $\max_y p(y\|x)$ | default | $.95 \pm .11$ | $.76 \pm .12$ | $.84 \pm .09$ | $\mathbf{.81} \pm .07$ |
| | SimCLR style | $\mathbf{.96} \pm .12$ | $.76 \pm .11$ | $.91 \pm .05$ | $.80 \pm .10$ |
| | no logits normalization | $.95 \pm .11$ | $.76 \pm .13$ | $.84 \pm .11$ | $.81 \pm .03$ |

Table 5: OOD Detection Results. The model is WideResNet-28-10 (without BN) following the settings of JEM (Grathwohl et al., 2019). The training dataset is CIFAR-10. Values are AUROC.

cross-entropy loss. The reason may be training with many negative examples helps to discriminate between positive and negative samples.

**OOD detection.** We evaluate HDGE with different value of $K$ by running experiments on the $\log p(x)$ based OOD tasks, we use the same experiments setting as Section 5.1.

We vary the batch size of SGLD sampling process in Grathwohl et al. (2019), effectively, we change the number of samples used to estimate the derivative of the normalization constant $\mathbb{E}_{p_\theta(x')}\left[\frac{\partial E_\theta(x')}{\partial \theta}\right]$ in the JEM update rule Equation (4). Specifically, we increase the default batch size $N$ from 64 to 128 and 256, due to running the SGLD process is memory intensive and the technique constraints of the limited CUDA memory, we were unable to further increase the batch size. We also decrease $K$ in HDGE to $\{64, 128, 256\}$ to study the effect of approximation.

The results are shown in Table 6, the results show that HDGE with a small $K$ performs fairly well except on CelebA probably due to the simplicity of other datasets. We note HDGE($K = 64$) outperforms

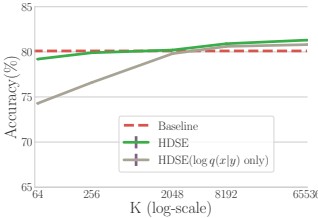

Figure 5: Accuracy comparison with respect to different $K$ on CIFAR-100. The baseline is standard cross-entropy loss. The model is WideResNet-28-10. Batch size is 256.

JEM and three out of four datasets, which shows the approximation in HDGE is reasonable good. While increasing batch size of JEM improves the performance, we found increasing $K$ in HDGE can more significantly boost the performance on all of the four datasets. We note JEM with a large batch size is significantly more computational expensive than HDGE, as a result JEM runs more slower than HDGE with the largest $K$.

| $s_\theta(x)$ | Model | Out-of-distribution | | | |
|---|---|---|---|---|---|
| | | SVHN | Interp | CIFAR100 | CelebA |
| | JEM ($N = 64$) (default) | .67 | .65 | .67 | .75 |
| | JEM ($N = 128$) | .69 | .67 | .68 | .75 |
| | JEM ($N = 256$) | .70 | .69 | .68 | .76 |
| $\log p(x)$ | HDGE ($K = 64$) | **.89** | **.79** | **.84** | **.62** |
| | HDGE ($K = 128$) | **.91** | **.80** | **.89** | **.73** |
| | HDGE ($K = 256$) | **.93** | **.81** | **.90** | **.76** |
| | HDGE ($K = 65536$) (default) | **.96** | **.82** | **.91** | **.80** |

Table 6: Ablation of approximation on detecting OOD samples. We use CIFAR10 for in-distribution. $N$ is the batch size of JEM and HDGE. HDGE uses $N = 64$. $K$ is the number of negative samples in contrastive learning.

