# OpenReview forum: "Hybrid Discriminative-Generative Training via Contrastive Learning"
_ICLR.cc/2021/Conference — Reject_

### Official Review · AnonReviewer1 · 2020-10-28

**Rating:** 3
**Confidence:** 5

**Review:**

===============Update after rebuttal period================
The connection between the contrastive learning objective and discriminative learning is made via "resemblance". And the author claims the "resemblance" as a theoretical contribution, which the first reason I vote for a clear rejection. This issue has not been addressed by the authors. The second reason for my rejection of the paper is the paper requires an effort to make it self-contained, especially for the experimental section. I remain my score of clear rejection.

=======================================================
This paper connected contrastive-learning and supervised learning from the perspective of the energy-based models. Then, the authors combine both objectives and evaluate the presented method on various datasets and tasks.

Strengths: The paper attempts to connect supervised and contrastive learning. I like the attempt. But unfortunately, I don't think it is valid. See explanations as follows.

Weakness:
1. I feel the claim in the paper is too strong. The approximation from equation 12 to 13 is very crude. Specifically, the approximation states that the infinite integral (for the normalization constant) can be replaced by a finite sum, which is generally not true.
2. Even if we assume the above approximation is fine, the connection with contrastive learning is very unclear. Precisely, the approximation is for modeling p(x|y), yet the contrastive learning is modeling p(x_1|x_2) with  x_1 and x_2 being the outcomes from correlated data. The authors do not discuss or compare between p(x|y) and p(x_1|x_2), and hence it makes the connection very vague.
3. The resulting objective (eq. 15) is a combination of the discriminative and generative modeling, which has already been studied.
4. On page 4, "the representation captures important information between similar data points, and therefore might improve performance on downstream tasks." This sentence is super vague, and I can't understand what the "important information" is and why if we capture "this important information", we "may improve performance on downstream tasks." The author should spend time polishing the presentation.
5. The main complaint of the presentation is the overclaim for the experimental section. I understand the contents are too much, and hence the author must move some experimental sections into Appendix. The author claims that the proposed method is performed on adversarial modeling and generative modeling, while these two sections only appear in the Appendix. In the last few lines of page 5, the author seems to rush the remaining experimental sections into the Appendix and asks the reviewer/reader to read themselves. The author should spend time arranging the contents and make sure the paper is self-contained.

==================================
Summary of the reasons why I vote for rejection:
1. The main contribution of the paper by connecting supervised learning and contrastive learning is overclaimed. The approximation of the intractable normalization term is not appropriate. The connection with contrastive learning is not solid.
2. The paper doesn't seem to be ready for submission. The content is not organized well and some ambiguous wordings should be avoided.


[1] Representation Learning with Contrastive Predictive Coding by Oord et al.
[2] On Variational Bounds of Mutual Information by Poole et al.

---

> ### Author Response · Authors · 2020-11-15
> **Response to AnonReviewer1**
>
> Thank you for your feedback on our work.
>
>
> **Q**: **Does the contrastive approximation to the generative term make sense?**  “I feel the claim in the paper is too strong. The approximation from equation 12 to 13 is very crude. Specifically, the approximation states that the infinite integral (for the normalization constant) can be replaced by a finite sum, which is generally not true.”
> **A**: Yes, we found even with a modest number of negative examples, the hybrid model outperforms JEM on OOD detection, classification, and other tasks, as shown in the ablation study. While the constant is intractable to estimate in practice, our finite number of negative examples approximation works well empirically. We think the contrast part is more important and leave getting around the normalization constant as a future research direction *[Jozefowicz et al, Mnih et al]*. Similar situation happens to the widely used noise contrastive estimation.
>
> **Q**: **Discussion and comparison between the approximation for $p(x|y)$ and contrastive loss** “Even if we assume the above approximation is fine, the connection with contrastive learning is very unclear. Precisely, the approximation is for modeling $p(x|y)$, yet the contrastive learning is modeling $p(x_1|x_2)$ with $x_1$ and $x_2$ being the outcomes from correlated data. The authors do not discuss or compare between $p(x|y)$ and $p(x_1|x_2)$, and hence it makes the connection very vague.”
> **A**: The approximation of p(x|y) is indeed the SimCLR-like contrastive loss used in representation learning. To see this, recall that SimCLR-like contrastive losses pull together between anchor data representation x and its positive example (e.g. x’, a different view of data x), and push apart between anchor x and negative examples $x_i$. The SimCLR objective is $\min_\theta -E_{p_{\rm data}(x)} \left[\log \frac{\exp(h_\theta(x)^\top h_\theta(x'))}{\sum_{i=1}^K \exp(h_\theta(x)^\top h_\theta(x_i))}\right]$.  Our method pulls together between anchor one-hot / label smoothed label vector y and positive example x (data points that have label y), and pushes apart between anchor y and negative examples x_i (other data points). Our (contrastive loss) objective is $\min_\theta - E_{p_{\rm data}(x,y)} \left[\log \frac{ \exp(f_\theta(x)[y])}{\sum_{i=1}^K \exp(f_\theta(x_i)[y])} \right ]$. Comparing the two, our loss is in fact a contrastive loss that resembles the SimCLR loss but focuses on data points and supervisions. We have made the connection between log q(x|y) and contrastive loss more clear in the revision.
>
> **Q**: **What is the novelty of this proposed hybrid model over previous studies.** “The resulting objective (eq. 15) is a combination of the discriminative and generative modeling, which has already been studied.”
> **A**: The original hybrid study by *[Ng & Jordan, 2002]* was done in the (simpler) context of Naive Bayes and Logistic Regression. Our work can be seen as lifting this work into today's context of training deep neural net classifiers. Other related previous work such as *[Larochelle & Bengio, 2008]* considers hybrid training for RBMs, one key distinction is that our method doesn't need a computationally expensive MCMC to train a model but uses contrastive learning to approximate the generative loss. To the best of our knowledge, our work is the first one to show contrastive approximation to the generative loss in hybrid models, and empirically demonstrate its effectiveness in thorough experiments.
>
>
> **Q**: **Suggestions to make the paper self-contained.** “The main complaint of the presentation … make sure the paper is self-contained.”
> **A**: We have arranged the experiments to make the paper self-contained in the revision.
>
>
> [1] Mnih A, Teh YW. A fast and simple algorithm for training neural probabilistic language models. arXiv preprint arXiv:1206.6426. 2012 Jun 27.
>
> [2] Jozefowicz R, Vinyals O, Schuster M, Shazeer N, Wu Y. Exploring the limits of language modeling. arXiv preprint arXiv:1602.02410. 2016 Feb 7.
>
> [3] Larochelle H, Bengio Y. Classification using discriminative restricted Boltzmann machines. InProceedings of the 25th international conference on Machine learning 2008 Jul 5 (pp. 536-543).
>
> [4] Ng AY, Jordan MI. On discriminative vs. generative classifiers: A comparison of logistic regression and naive bayes. InAdvances in neural information processing systems 2002 (pp. 841-848).

---

### Official Review · AnonReviewer4 · 2020-10-28

**Rating:** 5
**Confidence:** 3

**Review:**

1, Summary of contribution:
This paper gives a unified view of contrastive learning and supervised learning through energy-based model (EBM) training. Specifically, it claims that the contrastive loss can approximate the log-likelihood of the energy-based model, and shows that the proposed objective works on par with or better than its counterparts.

2, Strengths and Weaknesses:
The proposed method of training energy-based models achieves good performance on various tasks (e.g. OOD) without SGLD which usually requires a lot of tricks to work well on high-dimensional data.  This is the strength of this paper, and the community might be able to leverage the idea like this to extend the application of EBM to other domains in the future.
Meanwhile,  as the authors admit themselves,  the approximations are not thoroughly justified. Whether the research is appropriately bridging the gap between EBM and contrastive learning is questionable. This is the weakness of the paper.

3, Recommendation:
Marginally below the acceptance threshold. I believe that their claimed connection to EBM is overstated.

4, Reasons for Recommendation:
There are several factors that make me question whether the model learned by their proposed method can be legitimately interpreted as a variant of EBM. To name a few,
(1) As the authors write themselves, the approximation in eq (13) is crude. While the summands in the denominator are sampled from probability distributions (data distribution), when in fact it is supposed to be integral with respect to Lebesgue measure.
(2) The algorithm seems to be introducing some mysterious trick that makes the model ignore the gradient with respect to negative samples and normalization factor. For some unstated reason, the algorithm is also normalizing the logits.

The sheer results look promising, and it seems that the method is succeeding in capturing some aspects of the data distribution. However, they do not necessarily justify the claimed connection between HDGE and EBM. It could be that HDGE is just succeeding in learning the support of the dataset.
In order for this paper to reach the standard of ICLR, I believe that the authors must revise the strength of their work and redesign their experiments as such.

I believe that the paper can be improved if, within the scope of revision, the author can provide more solid justifications for approximations and tricks used in the paper.

---

> ### Author Response · Authors · 2020-11-15
> **Response to AnonReviewer4**
>
> Thank you for your feedback on our work.
>
> **Q**: **Is the model legitimately a variant of EBM?** “The sheer results look promising, and it seems that the method is succeeding in capturing some aspects of the data distribution. However, they do not necessarily justify … their experiments as such.”  “As the authors write themselves, the approximation in eq (13) is crude. While the summands in the denominator are sampled from probability distributions (data distribution), when in fact it is supposed to be integral with respect to Lebesgue measure.”
> **A**: While being an approximation to energy-based models, it is empirically effective. As our experiments demonstrated, HDGE not only outperforms JEM but also outperforms state-of-the-art contrastive loss (see e.g Figure 1 and Table 1). We believe this provides the community an interesting alternative to SGLD training to obtain nice properties of EBMs without suffering from numerical instability.
>
>
> **Q**: **Why ignoring gradients with respect to negative samples and normalization factor and why normalizing the logits?**  “ The algorithm seems to be introducing some mysterious trick that makes the model ignore the gradient with respect to negative samples and normalization factor. For some unstated reason, the algorithm is also normalizing the logits.”
> **A**: Ignoring gradients with respect to negative samples is a practice to scale up the number of negative samples without exploding cuda memory. This is also shown to be helpful in contrastive representation learning *[Chen et al, He et al]*. We chose to implement our method this way to make sure the algorithm is accessible to more researchers as it does not require an expensive GPU. A simpler alternative implementation that needs more cuda memory is large batch size, similar to what SimCLR does. To illustrate this, we implemented $\log q(x|y)$ without using MoCo memory and using a large batch size instead. The corresponding PyTorch code is shown in Algorithm 2 and the results on OOD tasks are shown in Table 5, we found that HDGE (SimCLR style) works as well as HDGE on these OOD datasets, and even performs slightly better. This shows that HDGE is robust to implementation choices.  The normalization of logits is in fact completely optional, we normalized them to keep the implementation difference minimal and ensure an apple-to-apple comparison with supervised contrastive loss. We have conducted experiments to evaluate HDGE without logits normalization,  the results are shown in Table 5. We found that HDGE without normalization performs on par with HDGE.
>
>
>
> [1] He K, Fan H, Wu Y, Xie S, Girshick R. Momentum contrast for unsupervised visual representation learning. InProceedings of the IEEE/CVF Conference on Computer Vision and Pattern Recognition 2020 (pp. 9729-9738).
>
> [2] Chen T, Kornblith S, Swersky K, Norouzi M, Hinton GE. Big self-supervised models are strong semi-supervised learners. Advances in Neural Information Processing Systems. 2020;33.

---

> > ### Comment · AnonReviewer4 · 2020-11-19
> > **Response to the authors**
> >
> > Thanks for the feedback. I believe that problem is not about the sheer result itself, but about the presentation.
> > I still feel that the paper lacks enough “theoretical” justification to be presented as a research of EBM.  At this point yet, I believe that HDGE shall not be presented as a version of EBM; it appears that the connection of HDGE to EBM is a rather superficial one that is not to be counted as a key contribution. That being said,  I would like to keep my score as is.

---

### Official Review · AnonReviewer3 · 2020-10-29
**Simple yet effective method for improvement over JEM**

**Rating:** 6
**Confidence:** 3

**Review:**

The paper proposes HDGE - a simple method to improve over JEM. JEM is optimized using a combination of two terms:
$\log p(y|x) + \log p(x)$

The first term is optimized using the standard cross-entropy loss, while the second term is optimized using SGLD. Running SGLD chains in each iteration can cause instability. In HDGE, instead of optimizing $\log p(x)$, an approximation to the conditional density $\log p(x|y)$ is optimized. The idea is to approximate the normalization constant $Z(\theta)$ with an empirical averaging of energy functions over a large memory bank. This yields a simple objective to optimize. The benefit of using such an approximation is that this eliminates the need for running SGLD, thereby improving the stability of training.

The idea itself is simple and intuitive. Experiments show that HDGE consistently outperform / perform on-par with JEM on image classification, OOD detection and calibration.

Should we call this a contrastive objective? Im not super convinced if the objective of $\log p(x|y)$ can be called a contrastive objective. Because in contrastive losses, we always focus on pairs of samples, i.e., we contrast the representation of one sample to another, while the objective in this paper takes the form similar to cross-entropy loss instead. Should the loss be called something instead?

I would like the authors to have a discussion on the training stability of HDGE compared to JEM. It looks like HDGE would be more stable since we don't need to run SGLD, but this message should be made more clear as this is the most important improvement over JEM.

The performance improvement in table 1 is marginal. So, it is important to perform multiple runs and report mean and standard deviations to understand the statistical significance of the results.

Can HDGE be used for generative modeling? i.e., how do you sample from p(x)? Experiments in appendix show that HDGE can be used in combination with JEM for generative modeling, but this again requires running SGLD. Can HDGE be used in isolation for generative modeling tasks?

How does the performance compare with other SOTA metods for OOD detection and calibation? Some comparisons that could be done include (but not limited to): Ren et al., "Likelihood Ratios for Out-of-Distribution Detection", Padhy et al., "Revisiting One-vs-All Classifiers for Predictive Uncertainty and Out-of-Distribution Detection in Neural Networks", Morningstar et al. "Density of States Estimation for Out-of-Distribution Detection", etc.

---

> ### Author Response · Authors · 2020-11-15
> **Response to AnonReviewer3**
>
> Thank you for your positive comments and feedback on our work.
>
> **Q**: **Should the loss be called something instead?**  “Should we call this a contrastive objective? Im not super convinced if the objective of $ log ⁡ p ( x | y )$  can be called a contrastive objective. Because in contrastive losses, we always focus on pairs of samples, i.e., we contrast the representation of one sample to another, while the objective in this paper takes the form similar to cross-entropy loss instead. Should the loss be called something instead?”
> **A**: You are right, this loss is a bit different from contrastive loss (e.g. SimCLR) used in representation learning. SimCLR-like contrastive losses pull together between anchor data representation x and its positive example (e.g. $x’$, a different view of data x), and push apart between anchor x and negative examples $x_i$. The SimCLR objective is $\min_\theta -E_{p_{\rm data}(x)} \left[\log \frac{\exp(h_\theta(x)^\top h_\theta(x'))}{\sum_{i=1}^K \exp(h_\theta(x)^\top h_\theta(x_i))}\right]$.  Our method pulls together between anchor one-hot / label smoothed label vector y and positive example x (data points that have label y), and pushes apart between anchor y and negative examples x_i (other data points). Our contrastive loss objective is $\min_\theta - E_{p_{\rm data}(x,y)} \left[\log \frac{ \exp(f_\theta(x)[y])}{\sum_{i=1}^K \exp(f_\theta(x_i)[y])} \right ]$. Therefore the generative loss term is in fact a contrastive loss that resembles the SimCLR loss but focuses on data points and supervisions. We have made the connection between $log q(x|y)$ and contrastive loss more clear in the revision.
>
> **Q**: **Emphasise on the training stability over energy-based models.** “I would like the authors to have a discussion on the training stability of HDGE compared to JEM. It looks like HDGE would be more stable since we don't need to run SGLD, but this message should be made more clear as this is the most important improvement over JEM.” <br/>
> **A**: We have highlighted the training stability of HDGE compared to JEM in the revision. We additionally highlighted that JEM is limited to models with no batch normalization due to severe training instability but HDGE can train such models without any numerical issues.
>
>
> **Q**: **Report results with standard derivations.**  “The performance improvement in table 1 is marginal. So, it is important to perform multiple runs and report mean and standard deviations to understand the statistical significance of the results.”
> **A**: The results reported in initial submission are mean of 5 runs. We have included standard derivations for each method in the revision.
>
>
> **Q**: **Comparison with state-of-the-art OOD methods.** “How does the performance compare with other SOTA metods for OOD detection and calibation? Some comparisons that could be done include (but not limited to): Ren et al., "Likelihood Ratios for Out-of-Distribution Detection", Padhy et al., "Revisiting One-vs-All Classifiers for Predictive Uncertainty and Out-of-Distribution Detection in Neural Networks", Morningstar et al. "Density of States Estimation for Out-of-Distribution Detection", etc."
> **A**: As R2 also asked, we have conducted a comparison with *[Winkens et al, 2020]* which is the current state-of-the-art OOD detection method. *[Winkens et al, 2020]* proposes to pretrain using contrastive loss and then finetune with a joint supervised and contrastive loss, and shows the SimCLR loss improves likelihood-based OOD detection. The results are shown in Table 2. We found HDGE outperforms *[Winkens et al, 2020]* on OOD tasks despite not using label smoothing and LARS optimizers.
>
>
> **Q**: **Can HDGE in isolation be used for generative modeling tasks?** “Can HDGE be used for generative modeling? i.e., how do you sample from $p(x)$? Experiments in appendix show that HDGE can be used in combination with JEM for generative modeling, but this again requires running SGLD. Can HDGE be used in isolation for generative modeling tasks?”
> **A**: We ran SGLD to generate realistic images, similar to JEM. We will explore the direction of using HDGE alone for generative modelling tasks. Interestingly though, even when used for generating samples tasks, HDGE does not suffer from the instability of JEM.
>
>
> [1] Winkens J, Bunel R, Roy AG, Stanforth R, Natarajan V, Ledsam JR, MacWilliams P, Kohli P, Karthikesalingam A, Kohl S, Cemgil T. Contrastive training for improved out-of-distribution detection. arXiv preprint arXiv:2007.05566. 2020 Jul 10.

---

### Official Review · AnonReviewer2 · 2020-11-02
**a good submission empirically improving on top of prior hybrid EBM works**

**Rating:** 6
**Confidence:** 4

**Review:**

Summary
- Paper proposes Hybrid Discriminative Generative training of Energy based models (HDGE) which combines supervised and generative modeling by using a contrastive approximation of the energy based loss
- Approach shows this is better than baselines on various tasks like confidence calibration, OOD detection, robustness and classification accuracy


Clarity
- Overall well written paper. Figures and tables are informative and supplement the flow.
- Formatting error in figure 4 in appendix


Novelty
- Paper proposes a simple but unified view of contrastive training, generative and discriminative modeling - a nice, novel contribution with empirically strong results
- Gets rid of computationally expensive SGLD by using contrastive approximation which was a key limitation of prior energy based modeling work like JEM


Significance
- Results are compelling across a wide range of tasks over existing (EBM) baselines including calibration, robustness, OOD detection, generative modeling and classification accuracy


Questions/clarifications/comments
- “, Grathwohl et al. (2019) show the alternative class-conditional EBM p(y|x) leads to significant improvement in generative modeling while retain compelling classification accuracy” -> Not sure the JEM model is class conditional

- How alpha = 0.5 (the weighting chosen)? The details are not presented.

- Error bars are missing for the classification accuracy experiments in Table 1 which makes it hard to verify improvements especially wrt supervised contrastive loss method

- Detail on how classification accuracy is computed when using generative term in HDGE is missing? Is it a linear classifier on top of learned representations?

- “Prior work show that fitting a density model on the data and consider examples with low likelihood to be OOD is effective” -> Not completely true see https://arxiv.org/abs/1810.09136

- Please share exact details on how p(x) for OOD score is calculated
- Error bars again missing in Table 2

- “We find HDGE performs beyond the performance of a strong baseline classifier” - this is a strong statement as only for CIFAR10/Celeb A the gains of HDGE are clear

- Why was the Winkens et al, 2020 contrastive baseline not used here to compare in Table 2 - https://arxiv.org/abs/2007.05566?
- “HDGE is conceptual simple to implement, scalable, and powerful.” -> conceptually. Also scalability is a somewhat strong claim as the main datasets used here are CIFAR variants.

- Was HDGE + JEM experiments also performed for OOD detection?

- Legend in figure 4 should be “HDGE” not “HDSE”?

Overall good effort with seemingly good improvements over prior efforts on hybrid EBMs over a number of tasks. Main concern is lack of error bars which makes it hard to validate claims in certain cases.

---

> ### Author Response · Authors · 2020-11-15
> **Response to AnonReviewer2**
>
> Thank you for your positive comments and feedback on our work.
>
> **Q**: **Discussion of the results of likelihood-based score for OOD detection.** “Prior work show that fitting a density model on the data and consider examples with low likelihood to be OOD is effective” -> Not completely true see https://arxiv.org/abs/1810.09136”
> **A**: We have included a discussion of the results of likelihood-based OOD score. While *[Nalisnick et al, 2019]* demonstrates powerful neural generative models trained to estimate density $p(x)$ can perform poorly on OOD detection, often assigning higher scores to OOD data points (e.g. SVHN) than in-distribution data points (e.g. CIFAR10), HDGE successfully assigns higher scores only to in-distribution data points, as shown in the histograms in Figure 1.
>
> **Q**: **How to choose proper weight \alpha.**  “How alpha = 0.5 (the weighting chosen)? The details are not presented.”
> **A**: Since $\alpha = 0$ leads to a discriminative model and $\alpha=1.0$ leads to a generative model. We choose $\alpha=0.5$ in our hybrid model for simplicity.
>
> **Q**: **How to compute classification accuracy in HDGE($\log q(x|y)$) only?** “Detail on how classification accuracy is computed when using generative terms in HDGE is missing? Is it a linear classifier on top of learned representations?”
> **A**: One can use a linear classifier on top of the learned representation to evaluate the representation, as commonly done in contrastive representation learning works. In our experiments, for a fair comparison with JEM, we directly evaluate the standard classifier $f(x)$ hidden in $\log q(x|y)$ (see equation (15)).
>
> **Q**: **Include details on how to compute OOD scores.** “Please share exact details on how p(x) for OOD score is calculated”
> **A**: The likelihood is derived from the classifier by LogSumExp. We have included the exact procedure of computing OOD scores $p(x)$ has been added to the experiments details section in Appendix.
>
>
> **Q**: **Comparison with *[Winkens et al, 2020]* and the results of HDGE+JEM.** “Why was the Winkens et al, 2020 contrastive baseline not used here to compare in Table 2 - https://arxiv.org/abs/2007.05566?” “Was HDGE + JEM experiments also performed for OOD detection?”
> **A**: We have conducted the comparison with *[Winkens et al, 2020]*, the results are shown in Table 2. *[Winkens et al, 2020]* proposes to pretrain using contrastive loss and then finetune with a joint supervised and contrastive loss, and shows the SimCLR loss improves likelihood-based OOD detection. We found HDGE outperforms *[Winkens et al, 2020]* on OOD tasks despite not using label smoothing and LARS optimizers. <br/>
> We found adding JEM to HDGE only led to mirror improvement, the reason might be the generative term in HDGE is a good regularization for the classifier therefore adding $q(x)$ does not add too much further improvement.
>
> **Q**: **Report results with standard derivations.** “Error bars are missing for the classification accuracy experiments in Table 1 which makes it hard to verify improvements especially wrt supervised contrastive loss method” “Error bars again missing in Table 2”
> **A**: The results in the submission are averaged over 5 random runs. We have reported standard derivations over 5 runs for each method in the revision.
>
>
> [1] Nalisnick E, Matsukawa A, Teh YW, Gorur D, Lakshminarayanan B. Do deep generative models know what they don't know?. International Conference on Learning Representations (ICLR) 2019.
>
> [2] Winkens J, Bunel R, Roy AG, Stanforth R, Natarajan V, Ledsam JR, MacWilliams P, Kohli P, Karthikesalingam A, Kohl S, Cemgil T. Contrastive training for improved out-of-distribution detection. arXiv preprint arXiv:2007.05566. 2020 Jul 10.

---

### Decision · Program_Chairs · 2021-01-07
**Final Decision**

**Decision:**

Reject

**Comment:**

The paper proposes hybrid discriminative + generative training of energy-based models (HDGE) building on JEM. By connecting contrastive loss functions to generative loss, HDGE proposes an alternative loss function that reduces computational cost of training EBMs.

The reviewers agree that this is an interesting idea and that the empirical results look promising.
However, multiple reviewers raised concerns that the theoretical justification was incomplete and felt that some of the claims about the equivalence between the two, as well as some of the practical approximations introduced, need more justification.

I encourage the authors to revise the paper and resubmit to a different venue.